# Contextuality in Classical Physics and Its Impact on the Foundations of Quantum Mechanics

**DOI:** 10.3390/e23080968

**Published:** 2021-07-27

**Authors:** Fritiof Wallentin

**Affiliations:** International Center for Mathematical Modeling in Physics and Cognitive Sciences, Linnaeus University, 351 95 Växjö, Sweden; fritiof.persson@lnu.se

**Keywords:** Koopman–von Neumann formulation of classical mechanics, contextuality, double-slit experiment, quantum interference, foundations of probability

## Abstract

It is shown that the hallmark quantum phenomenon of contextuality is present in classical statistical mechanics (CSM). It is first shown that the occurrence of contextuality is equivalent to there being observables that can differentiate between pure and mixed states. CSM is formulated in the formalism of quantum mechanics (FQM), a formulation commonly known as the Koopman–von Neumann formulation (KvN). In KvN, one can then show that such a differentiation between mixed and pure states is possible. As contextuality is a probabilistic phenomenon and as it is exhibited in both classical physics and ordinary quantum mechanics (OQM), it is concluded that the foundational issues regarding quantum mechanics are really issues regarding the foundations of probability.

## 1. Introduction

Feynman [1] pointed out that the interference phenomenon as observed in the double-slit experiment (DSE) can be seen as a purely probabilistic phenomenon, as a violation of the formula of total probability (FTP). As one is in the DSE still measuring probabilities in the same way, i.e., as frequencies of occurrence, Feynman concluded that FQM constitutes a probability theory on par with the classical one but only with different rules of computation. By ’classical’ Feynman meant that the computational rules equivalent to those in the measure theoretic formulation due to Kolmogorov (KPM) [2] where satisfied. Hence Feynman referred to the result of DSE as FQM violating classical probability theory. However, Ballentine [3] and Koopman [4], though not dismissing the probabilistic origin of the interference phenomenon in the DSE, disagreed with Feynman regarding his claim about it allegedly corresponding to a violation of FTP and classical probability theory. They correctly pointed out that the violation was only seemingly as such, resulting only because of a naive application of FTP. To briefly explain, we consider the DSE in a more generic form referred to as double-slit-type experiments (DSTE). In a DSTE, two observables *A* and *B* are considered. For simplicity, *A* is assumed dichotomous with possible outcomes being $a_{i}$, i=1,2. These observables are measured in three different contexts of measurement $C_{1,2}$, $C_{1}$ and $C_{2}$, where a *context of measurement* of an observable is a specification of the physical conditions under which this observable is measured. By construction, each context $C_{i}$, i=1,2 is such that the probability distribution over *A*, as measured under the circumstances that defines $C_{i}$, satisfies
(1)$$P\left(A=a_{j}|C_{i}\right)=\delta_{i,j}.$$Based on the measured probability distributions one then calculates the following function/distribution over *b*,
(2)$$P\left(B=b|C_{1,2}\right)-P\left(B=b|C_{1}\right)P\left(A=a_{1}|C_{1,2}\right)+P\left(B=b|C_{2}\right)P\left(A=a_{2}|C_{1,2}\right),$$
where the analogous notation as in (Equation 1) has been applied for the different probability distributions. Now, DSE corresponds to the particular DSTE for which *A* is the observable of the slit passage and *B* the observable of where on the screen the particle hits, with the respective contexts being:
$C_{1,2}$Both slits are open.$C_{1}$Slit 1 is open while slit 2 is closed.$C_{2}$Slit 2 is open while slit 1 is closed.
In DSE the actual result is a non-zero (Equation 2), which is what constitutes the interference phenomenon. If the context is ignored in the notation, this looks like a violation of FTP. However, as Koopman and Ballentine correctly pointed out, there is in reality no violation of FTP, Feynman had only failed to take the contexts into account. Indeed, as Feynman had shown [1], classical mechanics cannot account for the DSE. So DSE is ’unclassical’ in that sense. However, in the purely probabilistic sense there is nothing unclassical about it. In this article it will be shown that this typical quantum phenomenon is present in CSM as well. The above argument demonstrates that there is not inherent contradiction in this result.

Note that Koopman and Ballentine only disagreed with Feynman regarding his use of the term ’classical’. Their argument did not dismiss that FQM provided computational rules for calculating the term (Equation 2). Hence their views can be combined, which will be done in this article. FQM will be seen as a framework for computing probabilities in which the context dependence of observables is explicitly taken into account. This view of FQM will be made manifest by considering in the framework of *contextual probability*, as developed by Khrennikov in [5]. In it, violations of the type (Equation 2) are viewed as measures of contextuality. For a certain class of such violation FQM serves as a particular framework into which they can be represented. More specifically, this is done by considering the generic DSTE together with the *generalized formula of total probability* (generalized FTP), defined as
(3)$$\begin{array}{c} P\left(B=b|C_{1,2}\right)=P\left(B=b|C_{1}\right)P\left(A=a_{1}|C_{1,2}\right)+P\left(B=b|C_{2}\right)P\left(A=a_{2}|C_{1,2}\right)\\ +2\lambda_{b}\sqrt{P\left(B=b|C_{1}\right)P\left(A=a_{1}|C_{1,2}\right)P\left(B=b|C_{2}\right)P\left(A=a_{2}|C_{1,2}\right)} \end{array},$$
where the *interference coefficient* $\lambda_{b}$ serves as a measure of contextuality. A trivial interference coefficient means no contextuality is present in the particular DSTE. On the other hand, if
(4)$$|\lambda_{b}|\leq 1,$$
then we have contextuality. In [5], it shown that if a DSTE demonstrates such interference coefficients (Equation 4), then it can be represented in FQM. Indeed, in this case, the observables’ *A* and *B* are represented as mutually non-commuting self-adjoint operators. This will also be demonstrated here in Section 2. The purpose of doing so is to show that the occurrence of contextuality is, in FQM, equivalent to being able to physically tell the difference between pure and mixed states. The main result of Section 2 is that the way in which probability distributions transform under time evolution can be used to distinguish between mixed and pure states. It is in this sense that contextuality will be shown to occur in CSM.

In Section 3, KvN [6,7,8] will be presented and be shown to correspond to CSM. This will be in the sense of all solutions of the Schrödinger equation of KvN via Born’s rule also being solutions to the classical Liouville equation. As such, we can apply the statistical mechanical notion of an equilibrium state as states that are stationary in time and non-equilibrium as states not being stationary. In particular, the states of equilibrium are identified as corresponding to eigenvectors $\psi_{n}$ of the Liovillian $\hat{T}$, i.e., the KvN generator of time, as the associated probability distributions to these transform trivially under time evolution. The associated probability distributions of non-trivial superpositions of such eigenstates,
(5)$$\sum_{n}c_{n}\psi_{n},$$
are non-stationary. Hence, they can be identified as non-equilibrium states. By applying the principle of maximum Gibbs entropy, we will in fact be able to the main equilibrium states—the microcanonical and canonical ensembles—as corresponding to such eigenstates. Hence, it will be shown that KvN, when applying the principles of statistical mechanics, corresponds to CSM. As non-equilibirum states (Equation 4) transform non-trivially under time evolution while a corresponding mixed state
(6)$$\sum_{n}|c_{n}{|}^{2}|\psi_{n}\rangle \langle \psi_{n}|$$
does not, we are able to tell the difference between pure and mixed states. Hence, contextuality will be shown to be present in CSM, and is hence not only a phenomenon confined to OQM, where ’OQM’ refers to what is obtained through some variant of canonical quantization. In Section 4, it is shown what impacts on the foundation of quantum mechanics this has. In short, as contextuality is not restricted solely to OQM and as it is quantifiable as a purely probabilistic phenomenon, it is concluded that the issues regarding the foundations of quantum mechanics are really about the foundations of probability.

## 2. Time Evolution as an Indicator of Contextuality

Let *A* and *B* be two quantum observables on some Hilbert space H. For the sake of the argument, it suffices to assume that *A* is non-degenerate with an orthonormal eigenbasis ${\left\{\phi_{a}\right\}}_{a\in A}$, where the eigenvectors are labeled by their respective eigenvalue *a*. We impose no such restriction on *B*. We let $\psi_{b}$ denote an arbitrary (generalized) eigenvector of *B* with associated eigenvalue *b*. By Born’s rule, the probability distribution over *B*, given initial state ψ, is
(7)$$P\left(B=b|\psi \right):=|\langle \psi_{b},\psi \rangle {|}^{2}.$$
Analogously, for *A*,
(8)$$P\left(A=a|\psi \right):=|\langle \phi_{a},\psi \rangle {|}^{2}.$$
By utilizing the completeness relation in terms of ${\left\{\phi_{a}\right\}}_{a\in A}$, we obtain
(9)$$\begin{array}{cc} P(B=b|\psi ) & ={\left|\sum_{a\in A}\langle \psi_{b},\phi_{a}\rangle \langle \phi_{a},\psi \rangle \right|}^{2} \end{array}$$
(10)$$\begin{array}{cc} \; & =\sum_{a,a^{\prime }\in A}\langle \psi ,\phi_{a^{\prime }}\rangle \langle \phi_{a^{\prime }},\psi_{b}\rangle \langle \psi_{b},\phi_{a}\rangle \langle \phi_{a},\psi \rangle \end{array}$$
(11)$$\begin{array}{cc} \; & =\sum_{a\in A}P\left(B=b|\phi_{a}\right)P\left(A=a|\psi \right) \end{array}$$
(12)$$\begin{array}{cc} \; & +\sum_{a\neq a^{\prime }\in A}\langle \psi ,\phi_{a^{\prime }}\rangle \langle \phi_{a^{\prime }},\psi_{b}\rangle \langle \psi_{b},\phi_{a}\rangle \langle \phi_{a},\psi \rangle \end{array}$$
(13)$$\begin{array}{cc} \; & =\sum_{a\in A}P\left(B=b|\phi_{a}\right)P\left(B=b|\psi \right) \end{array}$$
(14)+2∑a<a′∈ARe{〈ψ,ϕa′〉〈ϕa′,ψb〉〈ψb,ϕa〉〈ϕa,ψ〉}.
Now, there exists a unique number θ∈[0,2π) such that
(15)$$\begin{array}{cc} \; & \langle \psi ,\phi_{a^{\prime }}\rangle \langle \phi_{a^{\prime }},\psi_{b}\rangle \langle \psi_{b},\phi_{a}\rangle \langle \phi_{a},\psi \rangle \\ \; & =\sqrt{P\left(A=a^{\prime }|\psi \right)P\left(B=b|\phi_{a^{\prime }}\right)P\left(B=a^{\prime }\right|\phi_{a}P\left(A=a|\psi \right)}e^{i\theta }. \end{array}$$
From which it follows that
(16)Re{〈ψ,ϕb′〉〈ϕb′,ψa〉〈ψa,ϕb〉〈ϕb,ψ〉}=P(A=a′|ψ)P(B=b|ϕa′)P(B=a′|ϕaP(A=a|ψ)cos(θb,a,a′),
with θ being unique if we restrict it to values in [0,π]. Hence, we end up with
(17)$$\begin{array}{cc} P(B=b|\psi ) & =\sum_{a\in A}P\left(B=b|\phi_{a}\right)P\left(A=a|\psi \right)\\ \; & +\sum_{a<a^{\prime }}\sqrt{P\left(A=a^{\prime }|\psi \right)P\left(B=b|\phi_{a^{\prime }}\right)P\left(B=a^{\prime }\right|\phi_{a}P\left(A=a|\psi \right)}\cos\left(\theta_{b,a,a^{\prime }}\right) \end{array}.$$
Hence, it has been shown that FQM indeed satisfies the generalized FTP (Equation 3).

Before moving on to the main point of this section, it is worth pointing out that we have implicitly interpreted quantum states as corresponding to contexts of measurement here. This is already evident from the choice of notation in (Equation 7) and (Equation 8). Note also that the meaning of a conditional probability—as the probability of outcome A=a given the conditions *C*—does by itself not necessitate that it must satisfy
(18)$$P\left(A=a|C\right)=\frac{P(A=a\;\mathrm{and}\;C)}{P(C)},$$
as it is defined in KPM, i.e., that *C* can be attributed to some random variable on the same measure space as *A*. Indeed, Kolmogorov himself [2] pointed out that probability measures
(19)P(·|C)
are based on the complex of all experimental conditions and that not all observables are representable as random variables on the same measure space. Hence, neither the notation in (Equation 7) and (Equation 8) nor the interpretation of quantum states as contexts cause any contradictions.

Now, assume that we have performed some DSTE from which we obtained the probability distributions
(20)$$\begin{array}{ccccccc} P(B=b|C_{1,2}), & \; & P(B=b|C_{1}), & \; & P(B=b|C_{2}) & \mathrm{and} & P(A=a|C_{1,2}) \end{array}$$
We wish to represent this DSTE in FQM. This means that *A* and *B* are to correspond to self-adjoint operators, which we, for simplicity, assume to be non-degenerate. As (Equation 1) holds by construction of the experiment it also means that $C_{1}$ and $C_{2}$ are to be represented as eigenvectors $\phi_{a_{1}}$ and $\phi_{a_{2}}$ of *A*, respectively. In addition, we must have
(21)$$P\left(B=b|C_{i}\right)={\left|\langle \psi_{b},\phi_{a_{i}}\rangle \right|}^{2}.$$
Furthermore, on the basis of $P(B=b|C_{1,2})$ alone, $C_{1,2}$ may be represented as
(22)$$\psi =\sum_{a\in A}e^{i\theta_{a}}\sqrt{P(A=a|C_{1,2})}\phi_{a},$$
up to some choices of the $\theta_{a}$ in [0,2π), or as the mixed state
(23)$$\rho =\sum_{a\in A}P\left(A=a|C_{1,2}\right)|\phi_{a}\rangle \langle \phi_{a}|.$$
However, as
(24)$$\left\{\begin{array}{c} P\left(B=b|\psi \right)={\left|\langle \psi_{b},\psi \rangle \right|}^{2}\\ P\left(B=b|\rho \right)=\langle \psi_{b},\rho \psi_{b}\rangle \end{array}\right.,$$
we obtain via a straightforward calculation that
(25)$$\begin{array}{cc} \; & P(B=b|\psi )-P(B=b|\rho )\\ \; & =\cos\left(\theta_{a_{1}}-\theta_{a_{2}}+\theta_{b,a_{1},a_{2}}\right)\sqrt{P\left(B=b|C_{1}\right)P\left(A=a_{1}|C_{1,2}\right)P\left(B=b|C_{2}\right)P\left(A=a|C_{1,2}\right)}, \end{array}$$
where $\theta_{b,a_{1},a_{2}}$ is the number such that
(26)$$\begin{array}{cc} \; & \langle \psi_{b},\phi_{a_{1}}\rangle \langle \phi_{a_{1}},\psi \rangle \langle \psi_{b},\phi_{a_{2}}\rangle \langle \phi_{a_{2}},\psi \rangle \\ \; & =e^{i\theta_{b,a_{1},a_{2}}}\sqrt{P\left(B=b|C_{1}\right)P\left(A=a_{1}|C_{1,2}\right)P\left(B=b|C_{2}\right)P\left(A=a_{2}|C_{1,2}\right)}. \end{array}$$
As
(27)$$P_{\rho }\left(B=b|\rho \right)=\sum_{i=1,2}P\left(B=b|C_{i}\right)P\left(A=a_{2}|C_{1,2}\right),$$
by comparing (Equation 25) with (Equation 3), we see that contextuality means that we are able to differentiate between pure and mixed states.

Recall that in an identification of $C_{1,2}$ as ψ in (Equation 22), there is the ambiguity of the complex phases $\theta_{b}$. This ambiguity can be thought of as the identification being only up to a unitary transformation *W* of the form
(28)$$W\psi =\sum_{a\in A}e^{i(\theta_{a}+\omega_{a})}\sqrt{P(A=a|C_{1,2})}\phi_{a}.$$
As ρ transforms trivially under all such transformations, we obtain from (Equation 25) that
(29)$$\begin{array}{cc} \; & P(B=b|W\psi )-P(B=b|\rho )\\ \; & =\cos\left(\theta_{a_{1}}^{\prime }-\theta_{a_{2}}^{\prime }+\theta_{b,a_{1},a_{2}}\right)\sqrt{P\left(B=b|C_{1}\right)P\left(A=a_{1}|C_{1,2}\right)P\left(B=b|C_{2}\right)P\left(A=a|C_{1,2}\right)}, \end{array}$$
where we have defined
(30)$$\theta_{a_{i}}^{\prime }:=\theta_{a_{i}}+\omega_{a_{i}}.$$That is, this difference between how pure and mixed transform under such unitary transformations is equivalent to contextuality. The relevance of this here is that in the case where *A* corresponds to the generator of time evolution, then the time evolution $e^{iAt}$ is such a transformation *W*. The equilibrium states of CSM will be shown to in KVN correspond to eigenvectors of the generator of time evolution $\hat{T}$, and as there certainly exist states in CSM that transform non-trivially under time evolution, it will have been shown that contextuality is not just an OQM phenomenon but occurs in classical physics as well.

Note, it was not discussed above whether there exists mixed states diagonal with respect to some other basis than {|a〉} such that their respective probability distributions with respect to *B* transform identically under time evolution. In Appendix A it is shown that there are no such states.

As a side note, the ambiguity of the complex phases in representing contexts as quantum states. Has crucial foundational importance when it comes to FQM and should be seen as a feature rather as something redundant. For instance, in a previous article [9], the author demonstrated that Born’s rule can be proved rather than postulated by enforcing the probability to be such that it is invariant under certain such unitary transformations *W*. Its foundational importance as it relates to Born’s rule has also for instance been demonstrated elsewhere, e.g., [10].

## 3. The Koopman-von Neumann Representation of Classical Mechanics

Let N∈N be the number of considered particles. We are considering a phase space $P≃R^{6N}$ together with a set of fixed global canonical coordinates (p,q). Typically, these canonical coordinates are chosen such that *q* corresponds to the observable of the position in Cartesian coordinates and *p*, corresponding to its the conjugate momentum [11] associated with *q*, defined as
(31)$$p:=\frac{\partial L}{\partial \dot{q}},$$
where *L* is the given Lagrangian function. We note that *p* defined in this way does not always correspond to linear momentum, e.g., the charged particle moving in a magnetic field [12]. However, for all Hamiltonian functions of the typical form
(32)$$H\left(p,q\right)=\frac{p^{2}}{2m}+V\left(q\right)$$
*p* does correspond to linear momentum. Furthermore, solely in terms of the formalism of Hamiltonian mechanics, there is nothing special about this particular choice of canonical coordinates. This choice of a coordinate *q* and its conjugate momentum does, however, play a distinguished role in going from the Lagrangian picture of classical mechanics to the Hamiltonian one via a Legendre transformation [11]. Note that it is not even a necessity to interpret *q* as a coordinate of position. We will, however, stick to this conventional interpretation of *p* and *q* here and consider a Hamiltonian function *H* which in terms of them has the typical form (Equation 31).

To be explicit with the notation, we have defined
(33)$$\left(p,q\right):=\left(p_{1}^{1},p_{2}^{1},p_{3}^{1},\ldots ,p_{1}^{N},q_{2}^{N},p_{3}^{N},q_{1}^{1},q_{2}^{1},q_{3}^{1},\ldots ,q_{1}^{N},q_{2}^{N},q_{3}^{N}\right).$$
As we are assuming these to be global coordinates, we can without loss of generality assume that each $p_{i}^{n}$ corresponds to the canonical projection $\pi_{3(n-1)+1}$, where
(34)$$\pi_{i}\left(x_{1},\ldots ,x_{K}\right):=x_{k}.$$
Similarly, each $q_{i}^{n}$ is defined as the projection $\pi_{3(N+n-1)+1}$. The time evolution of this Hamiltonian dynamical structure is given by a Hamiltonian flow
(35)$$U:t\in R\mapsto U_{t}\in \mathrm{Symp}\left(P\right),$$
satisfying
(36)$$\left(-\partial_{q}H,\partial_{q}H,\right)={\left.\frac{d}{d\tau }\right|}_{\tau =0}U_{\tau }.$$

Now, KvN corresponds to the unitary representation on $L^{2}\left(P\right)$ corresponding to the action
(37)$${\hat{U}}_{t}\psi =\psi \circ U_{-t}.$$
for every ψ. This representation is constructed by first defining it according to (Equation 37) on the space of Schwartz functions ψ on P. This action (Equation 37) acts bijectively on the Schwartz space as all $U_{-t}$ are smooth diffeomorphisms. As the Schwartz space is dense in $L^{2}\left(P\right)$ we may uniquely linearly extend this action (Equation 37) to all of $L^{2}\left(P\right)$. Now, by Liouville’s theorem [13], we have
(38)$$dx_{t}dy_{t}=dxdy,$$
where
(39)$$\left(x_{t},y_{t}\right):=U_{-t}\left(x,y\right).$$
Hence,
(40)$$\begin{array}{cc} \int \bar{\left({\hat{U}}_{t}\psi \right)}\left(x,y\right)\left({\hat{U}}_{t}\phi \right)\left(x,y\right)dxdy & =\int \bar{\psi }\left(U_{-t}\left(x,y\right)\right)\phi \left(U_{-t}\left(x,y\right)\right)dxdy \end{array}$$
(41)$$\begin{array}{cc} \; & =\int \bar{\psi }\left(x_{t},y_{t}\right)\phi \left(x_{t},y_{t}\right)dx_{t}dy_{t} \end{array}$$
(42)$$\begin{array}{cc} \; & =\int \bar{\psi }\left(x,y\right)\phi \left(x,y\right)dxdy. \end{array}$$Thus, each ${\hat{U}}_{t}$ preserves the inner product. $\hat{U}$ hence defines a unitary representation of the Hamiltonian flow *U*. Given this representation, we can make some natural identifications.

Consider the multiplication operators
(43)$$\left\{\begin{array}{c} \left({\hat{p}}_{i}^{n}\psi \right)\left(x,y\right)=x_{i}^{n}\psi \left(x,y\right)\\ \left({\hat{q}}_{i}^{n}\psi \right)\left(x,y\right)=y_{i}^{n}\psi \left(x,y\right) \end{array}\right.,$$
where we, similarly as in (Equation 33), have applied the notation
(44)$$\left(x,y\right):=\left(x_{1}^{1},x_{2}^{1},x_{3}^{1},\ldots ,x_{1}^{N},x_{2}^{N},x_{3}^{N},y_{1}^{1},y_{2}^{1},y_{3}^{1},\ldots ,y_{1}^{N},y_{2}^{N},y_{3}^{N}\right).$$
Since we have assumed that *U* acts on points (p,q) having the interpretation of kinetic momentum *p* and Cartesian position *q*, we naturally interpret the operator ${\hat{p}}_{i}^{n}$ as corresponding to the observable of the *i*th component of the kinetic momentum of the *n*th particle and ${\hat{q}}_{i}^{n}$ as corresponding to the observable of the *i*th component of the position of the *n*th particle. Based on this, the operator
(45)$$\hat{E}:=H\left(\hat{p},\hat{q}\right)$$
is interpreted as the observable of energy. We have here, for the sake of simplicity, dropped the sub-/supscripts, as will be done generally from now on unless their inclusion is a necessity.

As $\hat{U}$ defines a unitary representation of the Lie group R through *U*, we may apply Stone’s theorem to obtain a generator of time evolution $\hat{T}$. From (Equation 37), we see that $\hat{T}$ acts as
(46)$$\begin{array}{cc} \left(\hat{T}\psi \right)\left(x,y\right) & :={\left.i\frac{d}{d\tau }\right|}_{\tau =0}\left({\hat{U}}_{\tau }\psi \right)\left(x,y\right) \end{array}$$
(47)$$\begin{array}{cc} \; & =i{\left.\partial_{p}H\right|}_{\left(x,y\right)}\partial_{y}\psi \left(x,y\right)-i{\left.\partial_{q}H\right|}_{\left(x,y\right)}\partial_{x}\psi \left(x,y\right), \end{array}$$
on all $\psi \in L^{2}\left(P\right)$ in its domain D(T). Notice that we may also write $\hat{T}$ more concisely as
(48)$$\hat{T}\psi =i\left\{H,\psi \right\},$$
with {·,·} denoting the Poisson bracket on P, which in the coordinates (p,q), read
(49)$$\left\{f,g\right\}=\partial_{p}f\cdot \partial_{q}g-\partial_{q}f\cdot \partial_{p}g.$$The corresponding Schrödinger equation is, hence,
(50)$$i\partial_{t}\psi_{t}=\hat{T}\psi_{t}.$$In (Equation 48), $\hat{T}$ acts as a derivative. Hence, as $\hat{p}$ and $\hat{q}$ are multiplication operators, it follows that
(51)$$\begin{array}{cc} {}[\hat{T},\hat{p}]\psi & =i\left\{H,p\psi \right\}-i\hat{p}\left\{H,\psi \right\} \end{array}$$
(52)$$\begin{array}{cc} \; & =i\left\{H,p\right\}\left(\hat{p},\hat{q}\right)\psi +i\hat{p}\left\{H,\psi \right\}-i\hat{p}\left\{H,\psi \right\} \end{array}$$
(53)$$\begin{array}{cc} \; & =i\left\{H,p\right\}\left(\hat{p},\hat{q}\right)\psi \end{array}$$
(54)$$\begin{array}{cc} \; & =-i\partial_{q}V\left(\hat{p},\hat{q}\right)\psi . \end{array}$$That is,
(55)$$\left[\hat{T},\hat{p}\right]=-i\partial_{q}V\left(\hat{p},\hat{q}\right),$$
and similarly that
(56)$$\left[\hat{T},\hat{q}\right]=-i\frac{\hat{p}}{m}.$$
Indeed, these are merely the equations of motion in operator form. Hence, $\hat{p}$ and $\hat{q}$ satisfy the expected dynamical relations in line with their interpretation.

Next, we move on to the relation between the KvN Schrödinger equation and the Liouville equation. By Born’s rule,
(57)$$\rho_{t}:={\left|\psi_{t}\right|}^{2}$$
is a probability distribution over phase space. Assuming that $\psi_{t}$ is a solution to (Equation 50), it follows that
(58)$$\begin{array}{cc} \partial_{t}\rho_{t} & =\psi_{t}\bar{\partial_{t}\psi_{t}}+\bar{\psi_{t}}\partial_{t}\psi_{t} \end{array}$$
(59)$$\begin{array}{cc} \; & =i\psi_{t}\bar{i\partial_{t}\psi_{t}}-i\bar{\psi_{t}}i\partial_{t}\psi_{t} \end{array}$$
(60)$$\begin{array}{cc} \; & =i\psi_{t}\bar{\hat{T}\psi_{t}}-i\bar{\psi_{t}}\hat{T}\psi_{t} \end{array}$$
(61)$$\begin{array}{cc} \; & =i\psi_{t}\bar{i\{H,\psi_{t}\}}-i\bar{\psi_{t}}i\left\{H,\psi_{t}\right\} \end{array}$$
(62)$$\begin{array}{cc} \; & =\psi_{t}\bar{\left\{H,\psi_{t}\right\}}+\bar{\psi_{t}}\left\{H,\psi_{t}\right\} \end{array}$$
(63)$$\begin{array}{cc} \; & =\{H,\rho_{t}\}. \end{array}$$
This means that $\rho_{t}$ solves the Liouville equation. We, furthermore, note that if $\psi_{\lambda }$ is an eigenvector of $\hat{T}$, i.e.,
(64)$$\hat{T}\psi_{\lambda }=\lambda \psi_{\lambda },$$
then the probability distribution
(65)$$\rho_{\lambda }:={\left|\psi_{\lambda }\right|}^{2}$$
satisfies
(66)$$\begin{array}{cc} \left\{H,\rho_{\lambda }\right\} & =\psi_{\lambda }\bar{\left\{H,\psi_{\lambda }\right\}}+\bar{\psi_{\lambda }}\left\{H,\psi_{\lambda }\right\} \end{array}$$
(67)$$\begin{array}{cc} \; & =i\psi_{\lambda }\bar{\hat{T}\psi_{\lambda }}-i\bar{\psi_{t}}\hat{T}\psi_{\lambda } \end{array}$$
(68)$$\begin{array}{cc} \; & =i\psi_{\lambda }\bar{\lambda \psi_{\lambda }}-i\bar{\psi_{t}}\lambda \psi_{\lambda } \end{array}$$
(69)$$\begin{array}{cc} \; & =0. \end{array}$$
We hence see that eigenvectors of $\hat{T}$ correspond to statistical equilibria, while non-trivial superpositions of them correspond to statistical non-equilibria as their induced probability distribution, via Born’s rule, transforms non-trivially under $\hat{T}$.

To recap, abstractly speaking, we have unitarily represented the Hamiltonian flow *U* as $\hat{U}$ such that there exist (non-trivial) operators $\hat{p}$ and $\hat{q}$ satisfying (Equation 55) and (Equation 56), with $\hat{T}$ being the associated generator of time evolution of $\hat{U}$. Note that OQM is also such a representation of *U*. As such, KvN and OQM are, at this level, equivalent. This equivalence is, however, broken when considering a specific representation of *U*. This representation theoretic view of the difference/similarity between KvN and OQM will, however, not be formalized further here. However, as KvN itself contains subrepresentations of $\hat{U}$ in which $\hat{p}$ and $\hat{q}$ still act invariantly and non-trivially, we will apply the essence of this representation theoretic view to identify these subrepresentations as ’proper’ quantum theories.

For instance, for every E>0, we can construct such a subrepresentation on the Hilbert space $L^{2}\left(P,\nu_{E}\right)$, where
(70)$$d\nu_{E}\left(x,y\right)=\delta \left(E-H\left(x,y\right)\right)dxdy,$$
which corresponds to a subspace of $L^{2}\left(P\right)$ in the sense of the direct integral [14], i.e.,
(71)$$L^{2}\left(P\right)≃\int_{\left[0,\infty \right)}^{⊕}L^{2}\left(P,\nu_{E}\right)dE.$$
As the measure $\nu_{E}$ is invariant under *U*, $\hat{U}$ is unitarily represented on each $L^{2}\left(P,\nu_{E}\right)$ in the same fashion as on $L^{2}\left(P\right)$. Of course, these representations $L^{2}\left(P,\nu_{E}\right)$ have the natural interpretation of corresponding to subrepresentations of KvN with fixed total energy *E*. The measure $\nu_{E}$ corresponds to the microcanonical measure as known from CSM.

Note that $\nu_{E}$ has not yet been identified as a quantum state and so cannot at this point be interpreted as a microcanonical ensemble for CSM. This identification will, however, come next. To do this, we first need to find eigenvectors to $\hat{T}$ in a ’generic enough’ fashion. We take this to mean that *U* can be assumed to be periodic with a period τ and that there exists a smooth function Ω on phase space such that
(72)$$\Omega \circ U_{-t}=\Omega +t.$$Then
(73)i{H,Ω}=it.Assuming, furthermore, that all conserved quantities are of the form g∘H, it then follows that a generic eigenvector of $\hat{T}$ is of the form
(74)$$\psi_{n,f}=e^{i\frac{2\pi }{\tau }n\Omega }f\circ H,$$
where n∈Z. Now, in $L^{2}\left(P\right)$, the only restriction of *f* is that it must be such that $\psi_{n}$ is in $L^{2}\left(P\right)$. Note, however, that *f* in the subrepresentations $L^{2}\left(P,\nu_{E}\right)$ simply corresponds to a constant, a normalization factor, and hence this degeneracy is removed. However, the degeneracy it represents regains physical meaning in the formalism of direct integrals. The degeneracy induced by the ambiguity of the choice of *f* can be considered as a section in the sense of direct integrals. That is, $\psi_{n,f}$ is identified as the element in
(75)$$\int_{\left[0,\infty \right)}^{⊕}L^{2}\left(P,\nu_{E}\right)dE$$
defined as
(76)$$\psi_{n,f}:E\mapsto \psi_{n,f(E)}:=e^{i\frac{2\pi }{\tau }n\Omega }f\left(E\right)\in L^{2}\left(P,\nu_{E}\right).$$
Now, the amplitude of $\psi_{n,f}$ as a section is given as
(77)$${\langle \psi_{n,f(E)},\psi_{n,f(E)}\rangle }_{E}.$$
Therefore, Born’s rule tells us that given the equilibrium state $\psi_{n,f}$, the probability distribution over the energy *E* is
(78)$$P\left(E|\psi_{n,f}\right)={\langle \psi_{n,f(E)},\psi_{n,f(E)}\rangle }_{E}={\left|f\left(E\right)\right|}^{2}\nu_{E}\left(P\right).$$
We can simply apply the principles of statistical mechanics to find the desired statistical equilibrium state here. For example, by maximizing the Gibbs entropy given a fixed mean energy, thus concluding that (Equation 78) must correspond to the canonical measure, i.e.,
(79)$${\left|f\left(E\right)\right|}^{2}\nu_{E}\left(P\right)=\beta e^{-\beta E}.$$
We can also maximize the Gibbs entropy given a fixed energy *E*. As
(80)$$P\left(p,q|\psi_{n,f(E)}\right)=\frac{\delta (E-H(x,y))}{\nu_{E}\left(P\right)}$$
is the only statistical equilibrium given a fixed energy, it is trivially the distribution that maximizes the Gibbs entropy. As such, we have applied the tools of statistical mechanics to provide the eigenstates of $\hat{T}$ physical interpretation as statistical equilibria. Now, as has already been pointed out, the statistical non-equilibria correspond to non-trivial superpositions of these equilibria. Hence, in accordance with Section 2, we have demonstrated that contextuality exists in CSM as well.

Note, if $\hat{T}$ commuted with both $\hat{p}$ and $\hat{q}$, then indeed no contextuality would be demonstrable by measuring their associated probability distributions or average as these would then be invariant under time evolution for all states. However, because of (Equation 55) and (Equation 56), this is not so for the relevant cases.

## 4. Conclusions

In Section 2, it was demonstrated that in FQM, the occurrence of contextuality is equivalent to being able to differentiate between pure and mixed states. In turn, it was shown that mixed states that are diagonal in the eigenbasis of the considered generator of time evolution transform trivially under time evolution while the corresponding pure states do not. Hence, one can, through time evolution, demonstrate contextuality.

In Section 3, KvN was developed as a unitary representation of the Hamiltonian flow *U* on $L^{2}\left(P\right)$ through the action (Equation 37). Similarly to OQM, it was noted that in KvN *U* is represented such that there exist self-adjoint operators $\hat{p}$ and $\hat{q}$ satisfying the operator version of Hamilton’s equations of motion (Equation 55) and (Equation 56), which can be seen as a necessity for interpreting $\hat{p}$ and $\hat{q}$ as, respectively, corresponding to quantum observables of momentum and position. As such, KvN and OQM can be seen as equivalent at this pre-representation level. This representation theoretic view was not formalized further. However, the heuristics of it were used to motivate us to consider subrepresentations of KvN for which this still held. In particular, the subrepresentations considered were those corresponding to constant energy, these were subrepresentations in the sense of the direct integral (Equation 71). It was also shown that solutions of the KvN Schrödinger equation via Born’s rule are also solutions of the Liouville equation. In particular, the eigenstates of the generator of time evolution of KvN were shown to, in the sense of statistical mechanics, correspond to statistical equilibria, while statistical non-equilibria correspond to non-trivial superpositions of them. The principle maximum entropy was applied to show that the equilibrium states in the subrepresentations of constant energy correspond to the microcanonical ensemble. It was also shown that maximizing the entropy given a fixed average energy gives the equilibrium states in the direct integral (Equation 71) that correspond to the canonical ensemble. It was shown that the eigenstates of the KvN generator of time evolution permit a physical interpretation as equilibria in the sense of CSM, and hence the generator of time evolution itself corresponds to an observable. As the difference between equilibrium and non-equilibrium is physically observable, it followed that when the principles of statistical mechanics are applied to KvN, KvN exhibits contextuality.

Now, a possible counterargument for this claim about CSM in the form of KvN exhibiting contextuality is that CSM does not need to be done in FQM, i.e., in the form of KvN. That is, however, besides the point. As explained in the introduction and in Section 2, FQM is merely a computational framework for calculating probabilities of outcomes. In FQM, contextuality takes the form of being able to distinguish between pure and mixed states. Contextuality is, however, still a more general probabilistic phenomena which is quantifiable via generalized FTP. Therefore, it does not matter which formalism is used as long as one is dealing with probability.

It is worth pointing out that it has not been claimed that KvN can replace OQM. KvN cannot, for instance, account for the DSE [7]. Hence, no such replacement can be done. However, no such replacement is needed either. What has been shown is that contextuality, which is often considered as a hallmark of the OQM phenomenon, exists in classical physics as well. As contextuality really is a probabilistic phenomenon, it is, in this article, concluded that the issues regarding the foundations of quantum mechanics are really about the foundations of probability. It is here worth pointing out that Kolmogorov himself pointed out that probabilities are inherently contextual [2].

One might argue against this conclusion of this article by claiming that all classical physics really is reducible to OQM, and that OQM is the ’more fundamental theory’. Based on this, one would then conclude that any contextuality demonstrated in CSM would in reality only be a result of, say, the contextuality induced by $\hat{p}$ and $\hat{q}$ satisfying the canonical commutation relations. However, classical physics does not so simply reduce to OQM through, say, some ℏ→0 limit [15]. Therefore, if one still insists on this reductionism, one would have to consider another form of reduction of a different mathematical form but still carrying the essence of the physical interpretation of ℏ→0, the existence of which it is far from obvious and quite possibly impossible. Therefore one cannot so simply disregard the contextuality in CSM demonstrated here as not a ’real’ quantum phenomenon. Moreover, this reductionistic view of classical mechanics as being reducible to OQM is common. It is often even applied on the larger landscape of physics, and the natural sciences in general. It is for instance claimed that thermodynamics is just statistical mechanics and that chemistry is just physics. However, as shown in [15,16], and also pointed out by Dyson [17], this type of reductionistic relation does not hold in general. In [15,16] concrete demonstrations of the failure of reductionism are presented. In these refererences and in [18] it is furthermore discussed what this non-reductionism implies for the hierarchical ordering of theories in terms of a contextified view of emergence versus reduction and ontic versus epistemic.

Lastly we note that Bell-violations may be seen as being induced by contextuality. For in the proof of Bell’s theorem [19] the validity of FTP is crucial, something which is even more obvious when considering it in its Wigner form [20]. As such, Bell violations are also ways of demonstrating contextuality [21]. Contextuality is really the hallmark phenomenon of quantum mechanics and, contrary to popular belief, it is present in classical physics as well.

## Data Availability

No new data were created or analyzed in this study. Data sharing is not applicable to this article.

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
