# Peer review of "Contextuality in Classical Physics and Its Impact on the Foundations of Quantum Mechanics"

_entropy, 2021, doi:10.3390/e23080968_

Round 1

Reviewer 1 Report

Wallentin studies a connection between quantum and classical mechanics in the Koopman-von Neuman (KvN) representation. An overall topic is very important and of an active research interest. Unfortunately, I cannot recommend a publication of the current manuscript since it basic premise is incorrect. 

The key point is that the author does not clearly state all the axioms of the KvN classical, thereby reaching incorrect conclusions. Additionally, Wallentin appears not be aware of the relevant literature on this topic since this is far from the first work discussing this question (and definitely not the last one!) I wish the author had carefully read chapter 1 of Ref. [4], which is readily accessible via https://arxiv.org/abs/quant-ph/0301172 . In particular, it is explicitly worked out in Sec. 1.5 why there is no interference effects in the double slit experiment within in the KvN mechanics.

Wallentin missed an important axiom of the KvN mechanics that states that physical observables are arbitrary real functions f(\hat{x}, \hat{p}) of the commuting operators of position and momentum, [\hat{x}, \hat{p}]=0. As a direct consequence, there are no interference effects in the KvN classical mechanics (see Sec. 1.4 of Ref [4] for details)! If this axiom is dropped than the KvN mechanics is physically not equivalent to the classical (Liouville) statistical mechanics. According to this axiom, Wallentin’s operator \hat{T} is not a valid observable, hence the reached incorrect conclusions. Note that in the literature \hat{T} is usually denoted by \hat{L}, which is called the Liouvillian generator of motion.

The fundamental difference between the KvN and quantum mechanics is value of the commutator between the position and momentum. In particular, it has been shown in [https://doi.org/10.1103/PhysRevLett.109.190403] that the Schrodinger equation can be derived from Wallentin’s Eq. (15) if the position and momentum obey the canonical computational relation, whereas the KvN equation of motion follows if the position and momentum commute.

Also the violation of Bell’s inequality can happen only if the algebra of observables is commutative, as was shown in the very first equation of Ref. [https://doi.org/10.1016/0375-9601(87)90075-2]. Hence, no violation of Bell’s inequalities in the KvN settings.

Section 4 of the current manuscript is mathematically sloppy and contains several mistakes. The author should consult, e.g., [https://doi.org/10.1103/PhysRevA.88.052108] for a consistent alternative. 

The very recent work [https://doi.org/10.1016/j.aop.2020.168090] is also very relevant to the current manuscript.

Author Response

"In particular, it is explicitly worked out in Sec. 1.5 why there is no interference effects in the double slit experiment within in the KvN mechanics."

Nowhere in the manuscript did I state otherwise. I literarily made that reference to state that KvN indeed cannot account for the double-slit experiment. The interference effect I am showing is of the same nature as the one in the double-slit experiment but not identical in its empirical manifestation. As I explained in the first two sections, quantum interference is a purely probabilistic phenomenon due to contextuality. I the revised manuscript I have tried to make this point even clearer and even included more references to this view. 

"Wallentin missed an important axiom of the KvN mechanics that states that physical observables are arbitrary real functions f(\hat{x}, \hat{p}) of the commuting operators of position and momentum, [\hat{x}, \hat{p}]=0. As a direct consequence, there are no interference effects in the KvN classical mechanics (see Sec. 1.4 of Ref [4] for details)!"

I checked that section before and I do not agree. For instance the calculation (1.4.4) is ill-defined as the considered eigenvector are only such in the generalised sense, i.e as distributions. They are non-normalizable. I also checked another article (https://arxiv.org/pdf/quant-ph/0105112.pdf) by the same author where a similar claim is made in eq. (3.34). The problem here is that this argument requires that such a choice always can be made. Indeed, given a complex valued function \psi, its value at each point x can always be written  as

\psi(x)=F(x) exp{iG(x)}     (1)

for some real numbers F(x) and G(x). Indeed, one can use this to define function F and G such that (1) holds point-wise. However, it is a non-trivial problem to show that such a construction always can be found such that G is differentiable, which it must in order for the argument to hold. 

In the manuscript, and even clearer in the revised manuscript, I show explicitly that the relative complex phases are critical even in KvN for without them all time-evolution is trivial.

I checked Koopman's original paper and could there not see any axiom stating that only the operators f(\hat{x}, \hat{p}) correspond to observables. But even if, that is besides the point. I provide the Liouvillian with physical meaning as an observable via the principles of statistical mechanics. Its eigenstates correspond to statistical equilibria. In the revised manuscript I have tried to make this even more concrete by showing how these can be identified as the typical microcanonical and canonical ensembles by further applications of the principles of statistical mechanics. I would also like to emphasise that it is not only the observables f(\hat{x}, \hat{p}) which have physical significance in statistical mechanics. What separates classical mechanics from statistical mechanics are the extra principles regarding the probability distributions satisfying Liouville's equation, e.g that stationary such correspond to equilibria, the principles of indifference and the principle of maximal entropy. These principles, though being of grave physical significance, do not have their origin as observables f(\hat{x}, \hat{p}). The principles are furthermore invariant of the dynamical system considered, e.g they can be applied just as well to quantum mechanics as to classical mechanics. I have applied them to classical mechanics in the form of KvN and thus shown that KvN thus becomes classical statistical mechanics. So I do not see the relevance of your critique.

"Hence, no violation of Bell’s inequalities in the KvN settings"

I have nowhere claimed that KvN violates the Bell inequalities in the sense of one being able to account for, say, the violations demonstrated in Aspect's experiment via KvN. What I claim, which is mathematically true, is that the validity of the formula of total probability is a necessity for proving Bell's inequality. Violations of the formula of total probability is the phenomena that is typically referred to as quantum interference (Most notably Feynman has pointed this out in his article 'The Concept of Probability in Quantum Mechanics'). The point being that it is indeed quantum interference that is the  quantum phenomenon. Here I suggest reading Khrennikov's work some of which are referenced in the manuscript.

'Section 4 of the current manuscript is mathematically sloppy and contains several mistakes'

As no concrete examples where given I asked my colleagues (mathematicians) to check it. None could find anything mathematically wrong with it. 

What I agree is sloppy is the interpretations/motivations for the mathematics. This is why I omitted this section in the revised manuscript. I will instead incorporate it into a separate manuscript in which I can present it with a proper build-up to it. Perhaps it was this that you thought were sloppy. otherwise pleas provide me some concrete examples so that I can correct for them.

Lastly, though I found that most your critique was either misplaced or built on misconception. I have in any case tried my best to account for them in the revised manuscript. At the very least sort out the misconceptions. 

Author Response

I have now revised my manuscript according to your critique. I have tried making it much more streamlined. In doing so I removed section 4, the part about quantisation as representation theory. I decided to turn that part into a separate manuscript for later submission. 

I am curios, in what way did the mathematical part be improved? I checked with my colleagues (mathematicians) they did not find anything wrong with it especially not with regards to physics standards. I do agree however agree that not enough motivations for the mathematics are presented there and that those that are there are unclear. This is the reason I am turning it into a separate manuscript. There is much there that can use some extra biuld up to it. 

Round 2

Reviewer 1 Report

First of all, I very much appreciate the author editing the paper. It is now clear for me what is being done. I have to confess I find the results pretty trivial and the presentation unnecessary long. I do not think it would be a useful contribution to the field. This is my personal judgment. But since I do not find an explicit mistake, I could offer an unenthusiastic acceptance.

Minor suggestions:

  • Why there different fonts for $P$ in Eqs. (2) and (3)?
  • Using many abbreviations significantly inhibits readability of any manuscript.

Author Response

Though it is unenthusiastic I still thank you for your acceptance. I have adjusted the paper according to your minor suggestions. 

However, I must say, with your previous critical comments in mind it does not seem to be the case that the results of the paper are trivial, at least not from your perspective. For instance, you claim that no interference/contextuality is present in KvN. I show that it in fact is present. As I see it, from your perspective this would hence seem like a non-trivial result of the highest kind as it is the opposite of what you expected. 

Reviewer 2 Report

I think after the revision the paper should be published in Entropy.

Author Response

I thank you for your approval. I have to the best of my ability corrected for the spelling errors in the newly uploaded revised version.